# *Streptococcus suis* serotyping by matrix-assisted laser desorption/ionization time-of-flight mass spectrometry

Chadaporn Chaiden[1], Janthima Jaresitthikunchai[2], Anusak Kerdsin[3], Nattakan Meekhanon[4], Sittiruk Roytrakul[2], Suphachai Nuanualsuwan[1,5]*

**1** Department of Veterinary Public Health, Faculty of Veterinary Sciences, Chulalongkorn University, Bangkok, Thailand, **2** Functional Proteomics Technology Laboratory, Functional Ingredients and Food Innovation Research Group, National Center for Genetic Engineering and Biotechnology, National Science and Technology for Development Agency, Pathum Thani, Thailand, **3** Faculty of Public Health, Kasetsart University Chalermphrakiat Sakon Nakhon Province Campus, Sakon Nakhon, Thailand, **4** Department of Veterinary Technology, Faculty of Veterinary Technology, Kasetsart University, Bangkok, Thailand, **5** Food Risk Hub, Research Unit of Chulalongkorn University, Bangkok, Thailand

* suphachai.n@chula.ac.th

## Abstract

*Streptococcus suis*, particularly *S. suis* serotype 2 (SS2), is an important zoonotic pathogen causing meningitis in humans worldwide. Although the proper classification of the causative and pathogenic serotype is salutary for the clinical diagnosis, cross-reactions leading to the indistinguishability of serotypes by the current serotyping methods are significant limitations. In the present study, matrix-assisted laser desorption/ionization time-of-flight mass spectrometry (MALDI-TOF-MS) analysis of extracted peptides was developed to improve the classification of serotype of *S. suis*. The peptide mass fingerprint (PMFs) database of *S. suis* was generated from the whole-cell peptides of 32 reference strains of *S. suis* isolates obtained from pigs. Thirty-two human *S. suis* isolates from clinical cases in Thailand were used to validate this alternative serotyping method in direct comparison to the multiplex (m) PCR approach. All reference strains, representing 32 serotypes of *S. suis*, exhibited their individual PMFs patterns, thus allowing differentiation from one another. Highly pathogenic SS2 and SS14 were clearly differentiated from the otherwise serologically closely related SS1/2 and SS1, respectively. The developed MALDI-TOF-MS serotyping method correctly classified the serotype in 68.8% (22/32) of the same serotype isolates generated from the PMFs database; while the validity for the clinical human isolates was 62.5% (20/32). The agreement between the MALDI-TOF-MS and mPCR serotyping was moderate with a Kappa score of 0.522, considering that mPCR could correctly serotype up to 75%. The present study demonstrated that PMFs from the developed MALDI-TOF-MS-based method could successfully discriminate the previously indistinguishable highly pathogenic SS2 and SS14 from SS1/2 and SS1, respectively. Moreover, this serotyping method distinguished pathogenic SS6, and so is an alternative approach of choice to rapidly and reliably serotype clinically pathogenic *S. suis* isolates.

**Data Availability Statement:** All relevant data are within the paper and its Supporting Information files.

**Funding:** This study was supported by The 100th Anniversary Chulalongkorn University Fund for Doctoral Scholarship; The 90th Anniversary of Chulalongkorn University Fund (Ratchadaphiseksomphot Endowment Fund); The Scholarship from the Graduate School, Chulalongkorn University to commemorate the 72nd anniversary of his Majesty King Bhumibol Adulyadej; The Agricultural Research Development Agency (Public Organization).

**Competing interests:** No authors have competing interests.

**Abbreviations:** MALDI-TOF-MS, matrix-assisted laser desorption/ionization time-of-flight mass spectrometry; PCA, principal component analysis; mPCR, multiplex polymerase chain reaction; PMFs, peptide mass fingerprints; SS, *Streptococcus suis* serotype.

## Introduction

*Streptococcus* (*S.*) *suis* is a zoonotic foodborne pathogen [1] that inhabits the nasal cavity and particularly the tonsils of pigs. People who work closely with live pigs can potentially become infected with this pathogen via open skin lesions leading to the clinical symptoms of septicemia, endocarditis, peritonitis, pneumonia, and meningitis [1, 2]. In the past, this bacterium has been classified by its capsular polysaccharide into up to 35 serotypes [1, 2], of which currently 29 serotypes are recognized [3, 4]. Of these, *S. suis* serotype (SS2) is the most significant serotype as it is predominantly associated with both diseased pigs and human clinical cases, and it has been reported as the causative agent in more than 70% of human clinical cases. Whereas SS14 and some other pathogenic serotypes are far less significant and account for only up to 3% of human clinical cases worldwide [5].

Although the prompt confirmation of the causative serotype as part of the clinical diagnosis is pivotal to the successful treatment rate of *S. suis* infection, more than 20% of human cases have gone undiagnosed due to inherent drawbacks in the current serotyping methods [5]. Initially, microbial identification of *S.suis* is presumed following biochemical tests. Subsequently, polymerase chain reaction (PCR) assay targeting the presence of the recombinant/repair protein gene *recN* [6] is then used to confirm *S. suis*. Serological methods have been considered as the standard procedures for serotyping *S. suis*. However, cross-reaction among pairs or groups of serotypes, such as SS1 and SS14; SS2 and SS1/2; SS2 and SS22; SS6 and SS16; and SS1, SS2, and SS1/2; can lead to the inconclusive or erroneous serotyping of *S. suis*. These indistinguishable serotypes are problematic since pathogenic serotypes in both pigs and humans, such as SS2, SS6, SS14, and SS16, are found among all the inconclusive pairs (or groups) of cross-reactions [5]. In addition, serotyping with all the typing antisera is laborious, time-consuming, and expensive, and preparing the antisera is difficult due to the high cost and labor associated with its production.

Even though multiplex (m)PCR has recently been developed and gained popularity as an alternative technique to serotype *S. suis* isolates, mPCR does not differentiate SS2 from SS1/2, or SS14 from SS1 due to the high capsular gene cluster similarity. So, additional or alternative methods are required to solve this issue. Lately, matrix-assisted laser desorption/ionization time-of-flight mass spectrometry (MALDI-TOF-MS) has been increasingly endorsed and adopted as an alternative approach to microbial identification [7], including *S. suis* identification [8, 9]. In this method, cellular proteins (and peptides) of *S. suis* are extracted and then used for species identification via peptide mass fingerprints (PMFs) formed using MALDI-TOF-MS.

Nevertheless, one previous study successfully examined few serotypes of SS using MALDI-TOF-MS still most of the serotypes were not classified especially undistinguishable serotypes by current serotyping methods [8], and so is tackled in this study. The peptide extraction method was modified from Bruker's recommendation [10], and the extracted peptides of *S. suis* were then analyzed by MALDI-TOF-MS. The aim of this study was to demonstrate *S. suis* serotyping by means of the modified peptide extraction method coupled with MALDI-TOF-MS, and in particular to differentiate those inconclusive pathogenic and non-pathogenic serotypes.

## Materials and methods

### Bacterial strains

In this study, 32 reference strains (SS1-14, SS16-20, SS22-32, SS34, and SS1/2) representing 32 different serotypes of *S. suis* (Table 1) along with local *S. suis* SS2 (*n* = 23) and SS14 (*n* = 9)

**Table 1. Repeatability of MALDI-TOF-MS serotyping of the reference *S. suis* serotypes after anaerobic culture.**

| Serotype | Strain | Source | MALDI-TOF-MS | | GenBank acc.no. of 16S rRNA gene |
|---|---|---|---|---|---|
| | | | Serotype (best match) | LSV | |
| 1 | NCTC10237 | Diseased pig | 1 | 2.187 | LR594043.1 |
| 2 | NCTC10234 | Diseased pig | 2 | 2.453 | LS483418.1 |
| 3 | 4961 | Diseased pig | 3 | 2.341 | AF009478.1 |
| 4 | 6407 | Diseased pig | - | - | AF009479.1 |
| 5 | 11538 | Diseased pig | - | - | AF009480.1 |
| 6 | 2524 | Diseased pig | 6 | 2.207 | AF009481.1 |
| 7 | 8074 | Diseased pig | 7 | 2.369 | AF009482.1 |
| 8 | 14636 | Diseased pig | 8 | 2.302 | AF009483.1 |
| 9 | 22083 | Diseased pig | 9 | 2.178 | AF009484.1 |
| 10 | 4417 | Diseased pig | - | - | AF009485.1 |
| 11 | 12814 | Diseased pig | - | - | AF009486.1 |
| 12 | 8830 | Diseased pig | - | - | AF009487.1 |
| 13 | 10581 | Diseased pig | 13 | 2.354 | AF009488.1 |
| 14 | 13730 | Diseased pig | 14 | 2.492 | AF009489.1 |
| 16 | 2726 | Diseased pig | 16 | 2.504 | AF009491.1 |
| 17 | 93A | Healthy pig | - | - | AF009492.1 |
| 18 | NT77 | Healthy pig | 18 | 2.539 | AF009493.1 |
| 19 | 42A | Healthy pig | 19 | 2.36 | AF009494.1 |
| 20 | 86–5192 | Diseased calf | 20 | 2.566 | AF009495.1 |
| 22 | 88–1861 | Diseased pig | - | - | AF009497.1 |
| 23 | 89–2479 | Diseased pig | - | - | AF009498.1 |
| 24 | 88-5299A | Diseased pig | 24 | 2.207 | AF009499.1 |
| 25 | 89-3576-3 | Diseased pig | 25 | 2.419 | AF009500.1 |
| 26 | 89-4109-1 | Diseased pig | - | - | AF009501.1 |
| 27 | 89–5259 | Diseased pig | 27 | 2.304 | AF009502.1 |
| 28 | 89–590 | Diseased pig | - | - | AF009503.1 |
| 29 | 92–1191 | Diseased pig | 29 | 2.3 | AF009504.1 |
| 30 | 92–1400 | Diseased pig | 30 | 2.256 | AF009505.1 |
| 31 | 92–4172 | Diseased calf | 31 | 2.124 | AF009506.1 |
| 32 | EA1172.91 | Diseased pig | 32 | 2.336 | AF009507.1 |
| 34 | 92–2742 | Diseased pig | 34 | 2.393 | AF009509.1 |
| 1/2 | 2651 | Diseased pig | ½ | 2.255 | AF009476.1 |
| Total | 32 | 32 | 22 (68.8%) | 2.34* | |

*Average LSV of correct MALDI-TOF-MS classifications.

isolates from humans were used, the strains were originally collected from hospital, distributed in 5 regions (north, northeast, central, east, and south) of Thailand. Sequencing of the 16S rRNA gene was used to confirm these 32 reference strains [11]. Bacteria were cultured on Columbia blood agar (Difco Laboratories, Detroit, Mich.) with 5% (v/v) sheep blood at 37°C in an anaerobic condition for 24 h. Reference strains of *Staphylococcus aureus* ATCC 25923 and *E. coli* DH5 alpha were included to study cross-reaction.

## Peptide extraction

Pure colonies of *S. suis* for peptide extraction were prepared on Columbia blood agar as above. Then ethanol was added to 70% (v/v) to precipitate out the bacterial peptides from the whole

bacterial cell. The bacterial suspension was centrifuged at 11,000 g for 5 min, and then the pellet was collected. Next, 5% (v/v) trifluoroacetic acid (TFA) in absolute acetonitrile (ACN) was added to the pellet and the suspension was dissolved by gentle vortex. The bacterial peptides dissolved in the supernatant were collected after clarification by centrifugation at 11,000 g for 5 min and then kept at -20˚C prior to MALDI-TOF-MS analysis. Lowry assay was used to quantify the concentrations of extracted peptides [12].

## Peptide analysis by MALDI-TOF-MS

The extracted peptides from the 32 reference strains of *S. suis* were mixed with the sinapinic matrix solution [sinapinic acid in 5% (v/v) TFA in absolute ACN] and then spotted onto the MALDI target plate and allowed to crystalize at room temperature before inserting the MALDI target plate into the MALDI-TOF-MS instrument. The mass spectrometry (MS) spectra were collected using an Ultraflex III TOF/TOF (Bruker Daltonik, GmbH) instrument in a linear positive mode with a mass range between 2–20 kDa. Five hundred shots were reiterated and accumulated with a 50 Hz laser per SS. Likewise, this MALDI-TOF-MS procedure was repeated for all 32 serotypes of *S. suis*. All MS spectra were analyzed for fingerprint spectra and subjected to PCA using the FlexAnalysis version 3.4 and ClinProTool version 3.0 software (Bruker Daltonik, GmbH). The ACTH fragment 18–39 (human), insulin oxidized B chain (bovine), insulin (bovine), cytochrome C (equine), and apomyoglobin (equine) were used as the external protein calibrations. Analysis of variance (ANOVA), Student's t-test and statistics in the software, were used to assess the statistically significant differences in the PMFs across bacterial strains.

**Generating of *S. suis's* PMFs database.** The *S. suis* PMFs database was generated by inserting 20 qualified MS spectra from each of the 32 individual reference serotypes into the MALDI Biotyper database system according to Bruker's recommendation. The PMFs of 32 serotypes in the database will be used as the references PMFs for the serotype characterization and to evaluate the repeatability, reproducibility and validation.

## Serotype analysis by mPCR

All reference strains of *S. suis* were serotyped by mPCR [13]. The results from the MALDI-- TOF-MS analysis were compared to those from the mPCR of the same samples in terms of the repeatability validation and agreement of tests. Four sets of primers targeting the capsule (*cps*) loci across 28 serotypes of *S. suis* were used. The first primer sets targeted SS1/2, SS1-3, SS7, SS9, SS11, SS14, and SS16; the second primer set targeted SS4, SS5, SS8, SS12, SS18, SS19, SS24, and SS25; the third primer set targeted SS6, SS10, SS13, SS17, SS23, and SS31; and the fourth primer set targeted SS21 and SS27-30. The mPCR serotyping of *S. suis* were followed as previously described [13].

## Repeatability

The same isolate of each serotype was repeatedly cultured and extracted in exactly the same manner. Eight spots from each serotype were applied to determine the precision of this developed serotyping method Bruker Biotyper. Log score values (LSVs) were used to evaluate the serotype classifications, where LSVs between 0–1.69 were merely reliable, between 1.70–1.99 were probable, and more than 2.00 were highly probable. Among 8 spots, the matched serotypes with highest LSVs will be considered as a correct classification.

## Validation

To assess the validity of this serotyping method against known clinical isolates of *S. suis*, 32 isolates of *S. suis* from human cases in Thailand, comprised of SS2 (*n* = 23) and SS14 (*n* = 9), were

cultured in the conventional anaerobic growth condition, extracted, and characterized by MALDI-TOF-MS as described earlier. Each individual isolate was spotted for eight entries. The serotype classification of the clinical isolates was evaluated by the same Bruker Biotyper LSVs and criteria.

## Reproducibility or culture-condition repeatability

To evaluate the repeatability of this serotyping method with serotypes cultured under a different condition, then the 32 serotypes of *S. suis* were cultured as before except in an aerobic growth condition, and then peptide extracted and serotype classified by MALDI-TOF-MS using exactly the same MALDI-TOF-MS protocol as described earlier. The serotype classification of was evaluated by the same Bruker Biotyper LSVs and criteria.

## Degree of test agreement

The 32 reference strains of *S. suis* were grown anaerobically or aerobically and subjected to both mPCR and developed MALDI-TOF-MS serotyping methods as described. The serotyping results are presented in a 2 × 2 contingency table of frequencies with the rows and columns representing the serotype results for both serotyping methods. The degree of agreement between these two methods was assessed using Cohen's kappa statistic ($\kappa$) [14], as shown in Eq (1);

$$\kappa = \frac{P_O - P_E}{1 - P_E},\tag{1}$$

where *n* is the total observed frequency, $O_D$ is the sum of observed frequencies along the diagonal (Table 2), $E_D$ is the sum of expected frequencies along the diagonal, $P_O$ is the $O_D/n$, and $P_E$ is the $E_D/n$.

## Results

### Peptide mass spectra of *S. suis*

After extraction, peptide masses between 2–20 kDa were collected and analyzed using the FlexAnalysis version 3.4 and ClinProTool version 3.0 software (Bruker Daltonik, GmbH). All of the extracted peptides delivered an adequate number and intensity (more than $10^4$ a.u.) of peptide mass. Extracted peptides from each individual serotype generated particular PMFs that contained their individual unique mass(es) and so differentiated each serotype from one another (Fig 1A–1D). Overall, 4420, 5337, 5965, 6634, 6748, 6834, and 8260 Da peptide masses were commonly found in most of the SSs. Interestingly, SS1/2, SS11, and, SS13, had some other unique masses (2990 and 3005 Da) in common.

The former SS32 and SS34 (*S. orisratti*) had a unique PMF pattern with the 4447, 6610, and 6775 Da peptide masses being common to these two serotypes and so Biotyper software classified SS32 and SS34 in another group, as shown in the dendrogram (Fig 2).

**Table 2. Contingency table of identification frequencies to evaluate *S. suis* serotypes using the MALDI-TOF-MS analysis of this study compared to mPCR test.**

| Observed | | Multiplex PCR | | |
|---|---|---|---|---|
| | | True serotype | False serotype | Sum |
| MALDI-TOF | True serotype | 59 | 3 | 62 |
| | False serotype | 19 | 22 | 41 |
| | Sum | 78 | 25 | 103 |

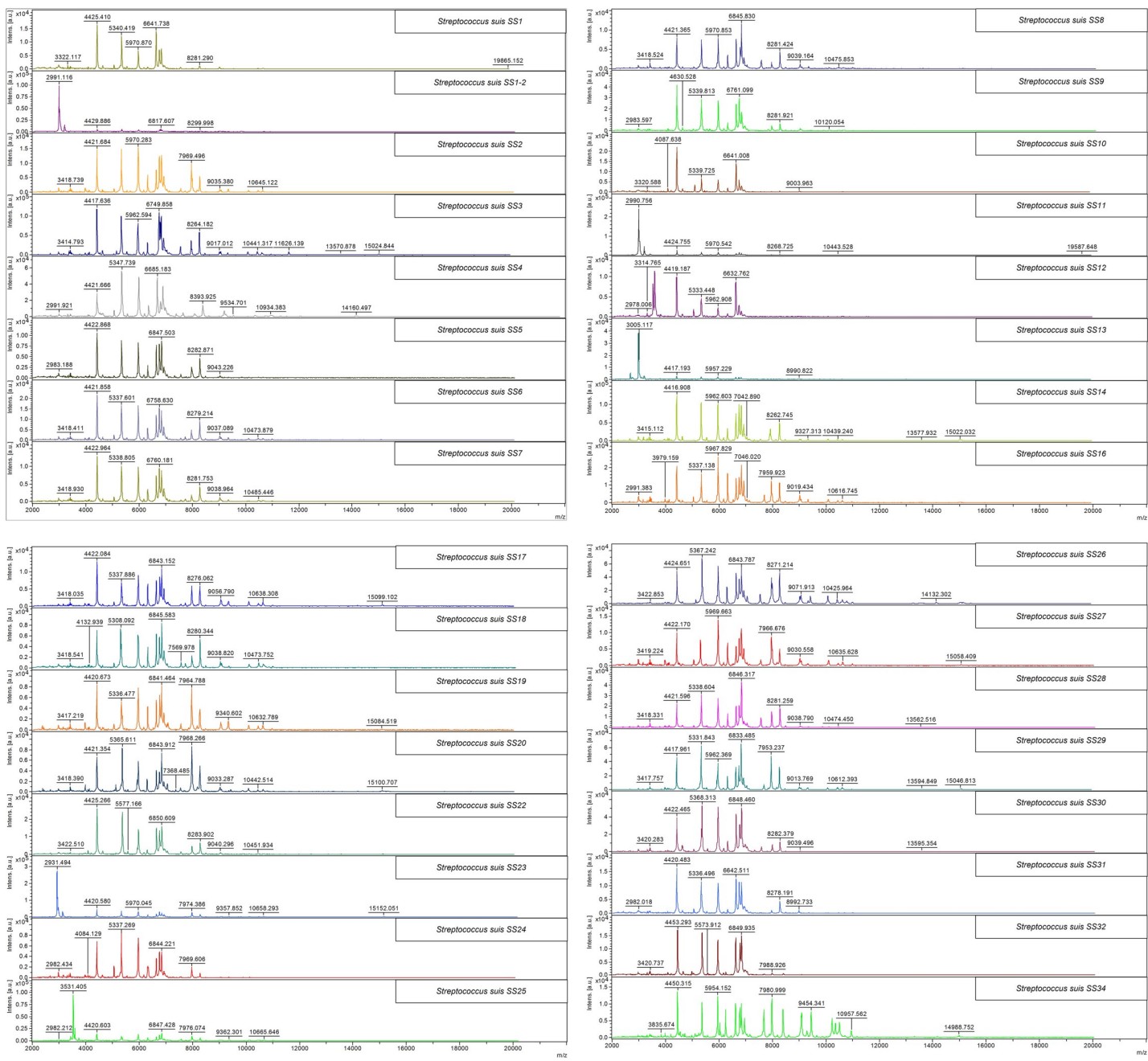

**Fig 1.** Representative MALDI-TOF peptide mass spectra of reference S. suis serotypes (A) SS1-7, (B) SS8-14 and SS16, (C) SS17-20 and SS22-25, and (D) SS26-32 and SS34. The X-axis represents the mass to charge ratio (m/z) and the Y-axis represents the intensity of the spectra.

The ambiguous serotypes by mPCR, which were the highly pathogenic SS2 and SS1 that are serologically associated with SS1/2 and SS14, respectively, were clearly distinguished from one another (Fig 1A and 1B). The 6956 Da peptide mass discriminated between SS2 and SS1/2 ($p < 0.01$), while the 6919 Da peptide mass discriminated between SS1 and SS14 ($p < 0.01$), as shown in Table 3. Likewise, principal component analysis (PCA) revealed that the PMFs of SS2, SS1/2, SS1, and SS14 distinctly clustered around their individual serotypes (Fig 3). Therefore, the PCA results did support that the ambiguity of serotype pairs SS2 and SS1/2; and SS1

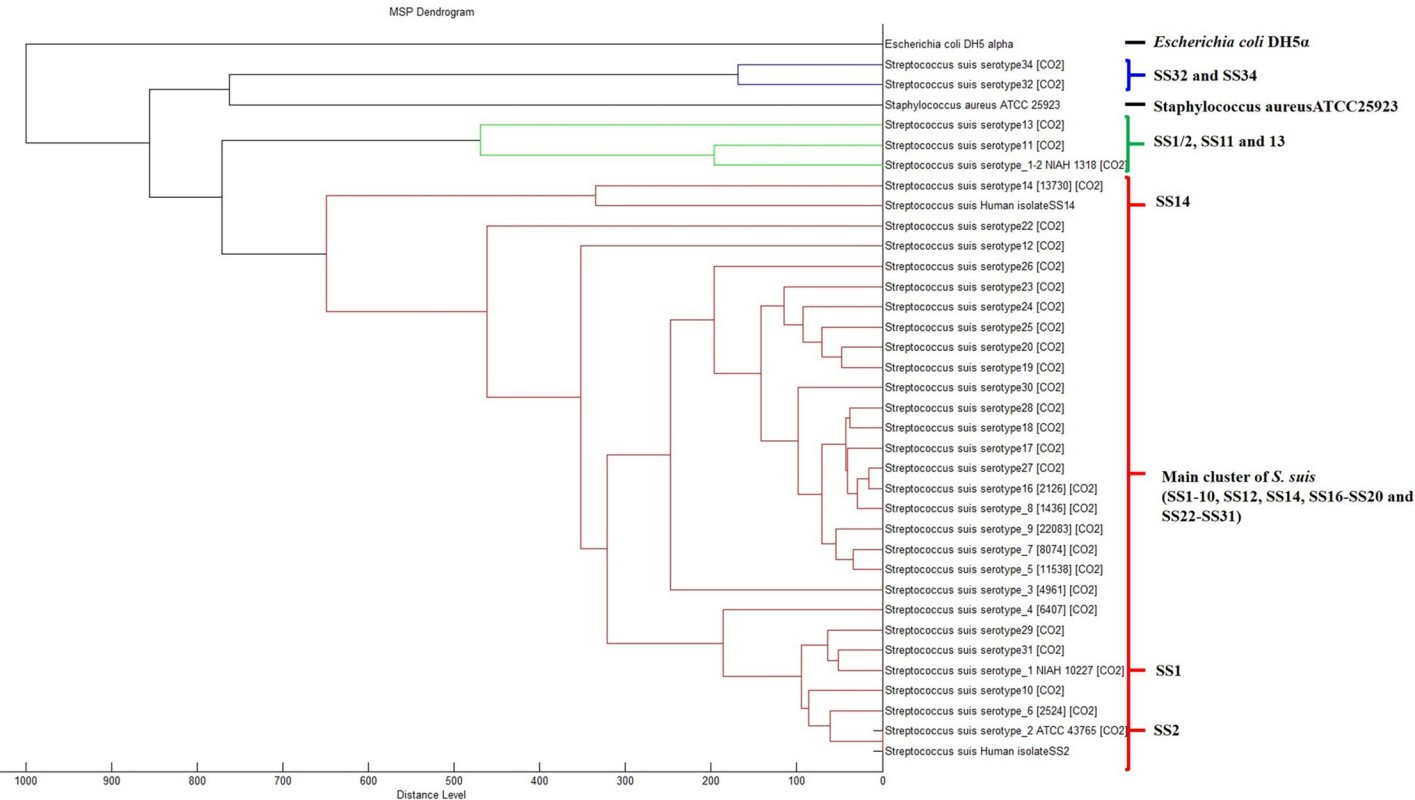

**Fig 2. Dendrogram based on the PMFs of reference SS1/2, SS1-14, SS16-20, SS22-32, SS34, and SS1/2 grown anaerobically.** SS32 and SS34 clustered in a group, while SS1/2, SS11, and SS13 clustered in another group and closer to the main SS PMFs. Staphylococcus aureus ATCC 25923 and E. coli DH5 alpha were included as the out group.

and SS14 could be clearly resolved by PMFs mapping using MALDI-TOF-MS. Additionally, cross-reacting serotypes based on serological methods were also classified, where the 6319, 6914, and 5056 Da peptides were index masses that differentiated between SS1 and SS2, between SS6 and SS16; and between SS2 and SS22, respectively, (Table 3).

## Repeatability

The same isolate of each serotype was repeatedly cultured, extracted, and classified in exactly the same manner to determine the precision of the developed serotyping method. Eight spots from each serotype were applied to classify comparing with the *S. suis* PMFs database

**Table 3. Differentiating between serologically ambiguous serotypes using the index mass obtained from MALDI-TOF-MS analysis of PMFs from anaerobically cultured *S. suis* serotypes.**

| Serotyping method | Ambiguous serotype | Differentiating index mass (Da) by MALDI-TOF-MS |
|---|---|---|
| Multiplex PCR | SS2 *vs.* SS1/2 | 6956 |
| | SS1 *vs.* SS14 | 6919 |
| Serological | SS1 *vs.* SS2 | 6319 |
| | SS2 *vs.* SS1/2 | 6956 |
| | SS6 *vs.* SS16 | 6914 |
| | SS2 *vs.* SS22 | 5056 |
| | SS1 *vs.* SS14 | 6919 |

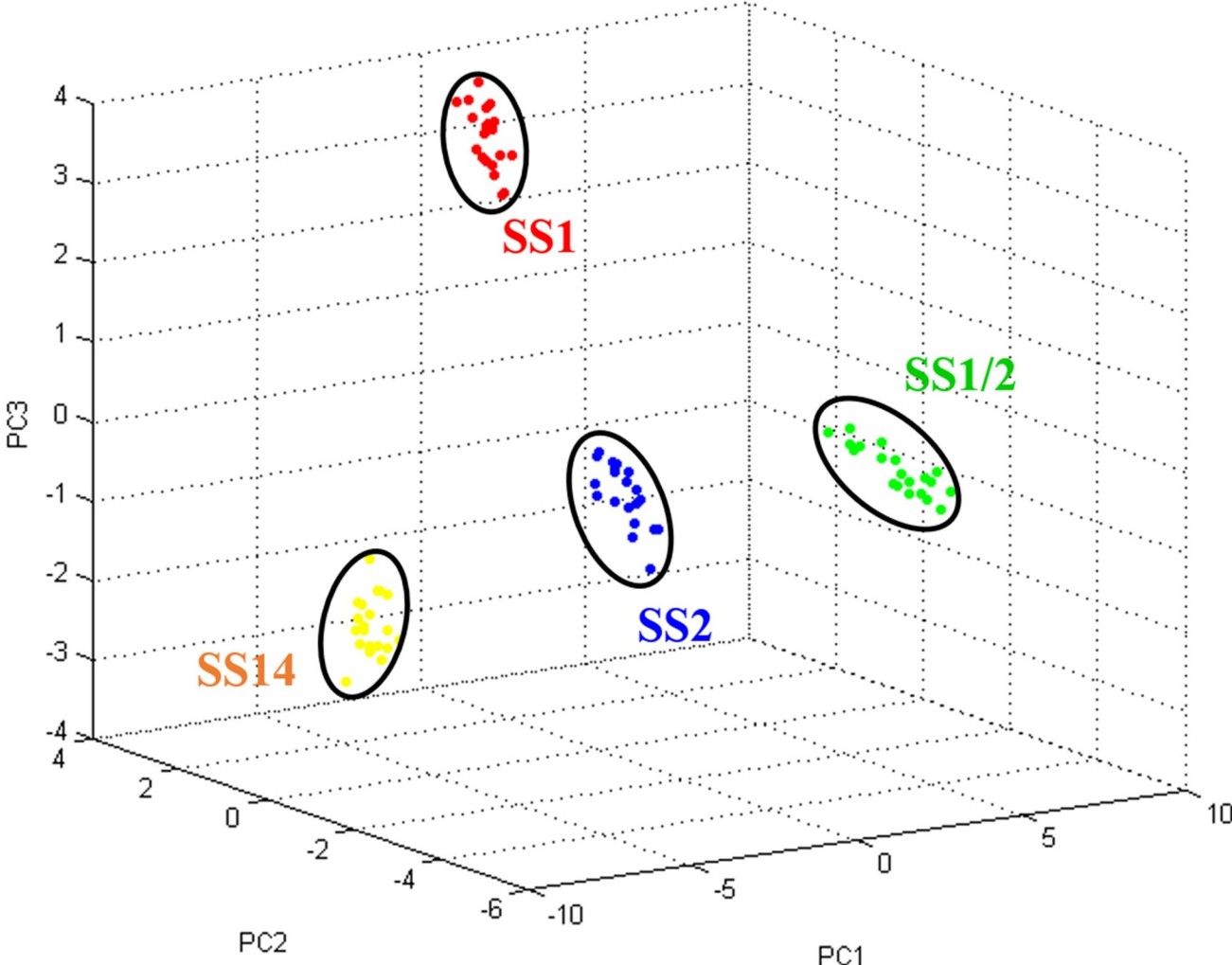

**Fig 3. The PCA, based on the PMFs, of the ambiguous serotypes of S.suis SS2, SS1/2, SS1, and SS14.** SS1, SS2, SS1/2, SS1, and SS14 represent the clusters of S. suis SS2, SS1/2, SS1, and SS14.

generated as described earlier. The results showed that 22 out of 32 serotypes (68.8%; average LSV of 2.34) were correctly serotype classified. The ambiguous serotypes by mPCR (SS2 with SS1/2, and SS1 with SS14) were also correctly serotyped with LSVs of 2.453, 2.255, 2.187, and 2.492, respectively. The serological cross-reacting serotypes SS1 with SS2, and SS6 with SS16, were correctly classified with LSVs of 2.187, 2.453, 2.207, and 2.504, respectively (Table 1). However, SS22 (cross-reactive with SS2) was unable to be serotyped (distinguished) by this MALDI-TOF-MS approach method.

## Validation

Thirty-two human isolates of *S. suis* from patients in Thailand, comprised of SS2 ($n = 23$) and SS14 ($n = 9$), were serotyped using the developed MALDI-TOF-MS method. The method correctly serotyped 20 out of 32 serotypes (62.5%) of human isolates, with an average LSV of 2.20 (Table 4). Serotype 2 was correctly classified in 13 out of 23 strains (56.5%, average LSV of 2.19) and SS14 was correctly classified in 7 out of 9 strains (77.8%, average LSV of 2.20).

**Table 4. Validation of MALDI-TOF-MS for serotyping *S. suis* human strains isolated in Thailand compared to the mPCR approach.**

| Serotype | Source | MALDI-TOF-MS | |
|---|---|---|---|
| | | Serotype (best match) | LSV |
| 2 | Human | 2 (13/23) | 2.19 |
| 14 | Human | 14 (7/9) | 2.20 |
| Total | 32 | 20 (62.5%) | 2.20* |

*Average LSV of correct MALDI-TOF-MS classifications.

## Reproducibility

Instead of using an anaerobic growth condition, 32 the serotypes of *S. suis* (SS1-14, SS16-20, SS22-32, SS34, and SS1/2) were cultured as before except under an aerobic growth condition. The strains were then peptide extracted and serotype classified by exactly the same MALDI-TOF-MS protocol. The MALDI-TOF-MS approach was found to correctly serotype 18 out of 32 strains of *S. suis* (56.3%) with average LSV of 2.34.

## Degree of test agreement

Cohen's unweighted Kappa statistic was used to elucidate the agreement between the mPCR and this developed MALDI-TOF-MS serotyping method, and gave a Kappa score of approximately 0.522. While the true serotype classifications of the MALDI-TOF-MS and mPCR were not statistically significant ($p > 0.05$) approximately 60% (95% CI: 51%-70%) and 76% (95% CI: 67%-84%), respectively.

## Discussion

As a rapid high throughput technique, MALDI-TOF-MS has been increasingly used in microbiological studies. It provides improved accuracy and power of resolution to identify or even classify microbial isolates [7, 15]. In this study, we modified the peptide extraction method coupled with MALDI-TOF-MS to identify *S. suis* and further classify the SSs. Mass spectra yielded from this peptide extraction technique exhibited good qualities, a high intensity, and an adequate number of peptide masses that overall indicated that this method could be applicable for SS classification.

The PMFs presented 4420, 5337, 5965, 6634, 6748, 6834, and 8260 Da peptide masses that were commonly found across all the tested SSs. Two of these masses were close to the 4420 and 8266 Da peptide masses that have recently been reported in previous studies using whole cell extraction of *S. suis*, which mass 4133 and 8367 were reported as the indicative species-specific peaks [8, 9]. This congruent finding may reach a general assumption that the 4420 and 8260 Da peptide masses could act as species-specific markers for *S. suis*. As these two masses were absent in others closely related *Streptococcus spp.* e.g. *S. plurextorum*, *S. porci* and *S.porcorum* where their species-specific markers corresponded to 6164, 6133 and 4190/8381, respectively [9]. However, some other peptide masses were found to be different from those in previous studies, which can be explained by the different preparation protocols, such as the acid concentration, spotting method, and type of matrices [16–18]. In the present study, the principal difference in the preparation protocol was the modified peptides extraction method, that formic acid was not included, gentle dissolve pellet by vortex and vigorously mix peptides with sinapinic matrix, which likely accounted for some of the dissimilar PMFs. Moreover, dissimilar PMFs could be inherent in the intraspecies discrepancy as well [19].

The PMFs of *S. suis*-like serotypes (SS32 and SS34) had a unique pattern with 4447, 6610, and 6775 Da peptide masses commonly found in these two serotypes. The 6775 Da peptide mass was somewhat close to the 6772 Da peptide mass that has previously been reported as a representative mass of *S. porcorum* PMFs [9]. Previous studies have suggested that SS32 and SS34 are likely to be *Streptococcus orisratti* [3]. In accord, the obtained PMFs dendrogram in this study (Fig 2) clustered both SS32 and SS34 together but separate from the other *S. suis* isolates, indicating that these two serotypes possibly possessed a high genetic dissimilarity from the other examined SSs. The phylogenetic tree based on 16s rRNA gene (Fig 4) also supported

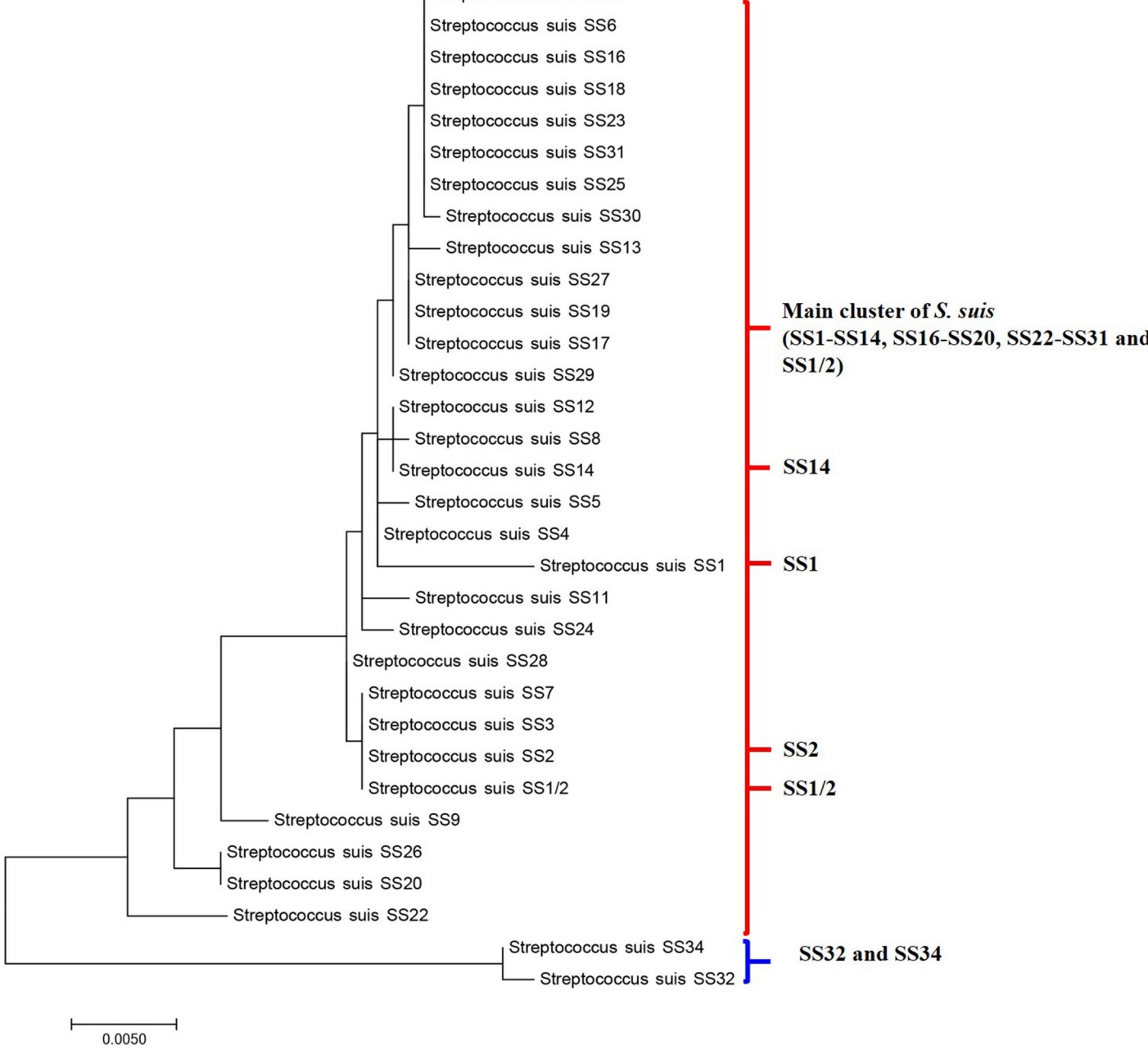

**Fig 4. Phylogenetic relationships of reference SS1/2, SS1-14, SS16-20, SS22-32, SS34, and SS1/2.** Maximum Likelihood method was used based on 16s rRNA gene.

this finding as SS32 and SS34 were also clustered separately from the main group of *S. suis*. Furthermore, SS1/2, SS11, and SS13 revealed some specific peptide masses that exclusively clustered together in the dendrogram (Fig 2). Thus, we presumed that these three reference serotype strains were divergent from the other tested reference serotype strains, in accord with a previous study that reported that SS13 was also divergent from other *S. suis* strains [20]. However, these divergent characters were not displayed in phylogenetic tree (Fig 4). In terms of the relatedness of PMFs, the cluster of SS1/2, SS11, and SS13 was closer to the main cluster than the cluster of SS32 and SS34 (Fig 2).

For SS classification, the PCA results demonstrated that the use of extracted peptides subject to MALDI-TOF-MS analysis was capable of discriminating between the high similarity serotypes pairs of (i) SS2 and SS1/2 and (ii) SS1 and SS14. Notably that, these two pairs of serotypes closely clustered together in phylogenetic tree (Fig 4) but separately in peptide dendrogram (Fig 2). This finding could be used to explained why our modified method could differentiate these two pairs of high similarity serotypes. Importantly, SS2 and SS14 are the two serotypes that accounted for most major *S. suis* infections [5, 21], where the misidentification of the causative pathogen could delay the appropriate treatment [22]. Moreover, the currently available classification methods are sometimes inconclusive. Hence, this modified peptide extraction method from Bruker's recommendation coupled with MALDI-TOF-MS analysis could be an alternative approach to discriminate between serologically cross-reactive SSs and so is a potentially significant advantage for *S. suis* infection diagnosis.

According to the repeatability test, the extracted PMFs coupled with our *S. suis* database was capable of discriminating 22 out of 32 (68.8%) reference SSs. Given that some serotypes of *S. suis* share the same structural components [23], then the structural peptides acquired by the present study could be similar and so have led to the misclassification of the other 10 serotypes (31.3%). Moreover, only 8 spots were applied for repeatability test, while 20 qualified spots were used to generate the *S. suis* PMFs database. Similarly, only 4 spots are sufficient to identify bacteria by means of MALDI-TOF-MS [15]. However, serotype classification is more subtle than bacterial identification thus requiring more replication of MS spectra to differentiate the more closely related serotype within the same species. In order to improve the correct serotype classification rate, we suggested that the repeatability test of the serotype classification, including unknown serotyping, should require at least 8 MS spectra (spots). Likewise, LSVs criteria of correct serotype classification are probable between 1.70–1.99. More PMFs included in the database from each individual serotype would warrant a higher degree of correct classification.

Clinical human isolates of *S.suis* were included in this study to validate this method. Just over half (56.5%) and three-quarters (77.8%) of human SS2 and SS14 strains, respectively, were correctly classified. This moderate level of validity of the serotyping method implied that the *S. suis* reference strains and field strains posed a somewhat dissimilar peptide background. Of relevance then is it has previously been reported suggested that different backgrounds of the same serotype of *S. suis* could result in different phenotype expressions [24]. For further study, more PMFs from field strains of *S. suis* should be included in the database to improve the reliability of the serotype classification.

The reproducibility or culture-condition repeatability of the serotype classification was merely acceptable (56.3%) when *S. suis* was cultured in an aerobic condition. The different background derived from different growth conditions of the same isolates could yield the dissimilar peptides, such as in this case, where the anaerobic *vs.* aerobic growth conditions of *S. suis* resulted in a mismatched classification. Therefore, the reproducibility of this serotyping method was growth-condition specific, as previously reported for *Burkholderia pseudomallei*, where altering the incubation condition lowered the identification scores of *Burkholderia pseudomallei* [18]. Therefore, various growth conditions, including the anaerobic growth of *S. suis*

should be recommended and emphasized when using this MALDI-TOF-MS classification coupled with corresponding PMFs database.

That only a moderate agreement between the MALDI-TOF-MS and mPCR analyses (unweighted Kappa score of 0.522) pointed out that the results among these methods were controversial. The mPCR is not without limitations, such as ambiguous serotype classifications corresponding to the moderate degree (76%) of the true serotype classification [13], while However, these divergent characters were not displayed in phylogenetic tree SS2 and SS14 from SS1/2 and SS1, respectively. The degree of agreement measured how well the two tests agreed with each other, but not on the reliability to correctly serotype *S. suis*. Nevertheless, 95% CI of both methods indicated that the true serotype classification rate of both methods was not statistically different. Theoretically, in order to improve the kappa score, more PMFs of *S. suis* should be included, since the accuracy of the classification depends on the number of reference spectra present in the database [25].

## Conclusions

In present study, we successfully differentiated the ambiguous serotypes of *S. suis*. Thus, providing an alternative method for *S. suis* classification that could be useful for *S. suis* infection diagnosis and epidemiological study of this zoonotic pathogen.

## Supporting information

**S1 Table. The complete list of m/z of *S. suis* serotyping by MALDI-TOF-MS identified and used in this study.**
(XLSX)

## Acknowledgments

We thank the staffs Center of Excellence for Emerging and Re-emerging Infectious Diseases in Animals, Chulalongkorn University, Bangkok, Thailand for supporting the PCR thermocycler and techniques.

## Author Contributions

**Conceptualization:** Sittiruk Roytrakul, Suphachai Nuanualsuwan.

**Data curation:** Chadaporn Chaiden, Janthima Jaresitthikunchai, Sittiruk Roytrakul.

**Formal analysis:** Chadaporn Chaiden, Janthima Jaresitthikunchai, Sittiruk Roytrakul.

**Funding acquisition:** Suphachai Nuanualsuwan.

**Investigation:** Chadaporn Chaiden, Janthima Jaresitthikunchai, Sittiruk Roytrakul.

**Methodology:** Janthima Jaresitthikunchai, Sittiruk Roytrakul.

**Resources:** Anusak Kerdsin, Nattakan Meekhanon.

**Supervision:** Janthima Jaresitthikunchai, Sittiruk Roytrakul.

**Validation:** Suphachai Nuanualsuwan.

**Visualization:** Chadaporn Chaiden.

**Writing – original draft:** Chadaporn Chaiden.

**Writing – review & editing:** Chadaporn Chaiden, Janthima Jaresitthikunchai, Anusak Kerdsin, Sittiruk Roytrakul, Suphachai Nuanualsuwan.

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
