## [Decision Letter · Decision Letter 0]

25 Feb 2021

PONE-D-21-02524

Streptococcus suis serotyp ing by matrix-assisted laser desorption/ionization time-of-flight mass spectrometry

PLOS ONE

Dear Dr. Nuanualsuwan,

Thank you for submitting your manuscript to PLOS ONE. After careful consideration, we feel that it has merit but does not fully meet PLOS ONE’s publication criteria as it currently stands. Therefore, we invite you to submit a revised version of the manuscript that addresses the points raised during the review process.

We look forward to receiving your revised manuscript.

Kind regards,

Joseph Banoub, Ph,D., D. Sc., FRSC

Academic Editor

PLOS ONE

Journal Requirements:

"We also thank the Agricultural Research Development Agency (Public organization), Bangkok, Thailand, for the financial support."

 "Chadaporn Chaiden

The 100th Anniversary Chulalongkorn University Fund for Doctoral Scholarship. The 90th Anniversary of Chulalongkorn University Fund (Ratchadaphiseksomphot Endowment Fund);

The Scholarship from the Graduate School, Chulalongkorn University to commemorate the 72nd anniversary of his Majesty King Bhumibol Adulyadej."

Reviewers' comments:

Reviewer's Responses to Questions

**Comments to the Author**

1. Is the manuscript technically sound, and do the data support the conclusions?

Reviewer #1: Yes

Reviewer #2: Yes

2. Has the statistical analysis been performed appropriately and rigorously? 

Reviewer #1: Yes

Reviewer #2: Yes

3. Have the authors made all data underlying the findings in their manuscript fully available?

Reviewer #1: No

Reviewer #2: Yes

4. Is the manuscript presented in an intelligible fashion and written in standard English?

Reviewer #1: Yes

Reviewer #2: Yes

5. Review Comments to the Author

Reviewer #1: In the manuscript titled “Streptococcus suis serotyping by matrix-assisted laser desorption/ionization time-of-flight mass spectrometry”, Chaiden et al. used MALDI-TOF mass spectrometry to serotype Streptococcus suis from reference samples and compare against the use of multiplex-PCR, then validate the mass spectrometry method using clinical isolates. Specific comments are as follows:

1. On line 83, it is a bit misleading to state that “the classification of SSs has never been reported”. In reading through reference #8, it seems the authors of that paper have been able to examine serotypes 2, 7, and 9, as well as what they referred to as “less prevalent” serotypes. Though not a complete listing of serotypes (as it seems the authors assessed at least eight serotypes), the authors of the current manuscript cannot state that any classification/serotype attempts have not been reported previously. This needs to be corrected through the manuscript, where appropriate.

2. Is multiplex-PCR really considered a “gold standard”, when it has a reasonably poor rate of identifying serotypes? Sequencing, and likely of just the 16S RNA, would really provide near 100% serotype identity. (After all, the authors in the current manuscript indicate that the 32 reference strains were determined by sequencing of the 16S rRNA). It is strongly recommended to remove reference to “gold standard” throughout the manuscript regarding multiplex-PCR.

3. Under “Validation”, provide the source/location of the clinical isolates used (i.e., which hospital or clinic, and what parts of the country the subjects came from).

4. Figure 2 is not clear at all at the resolution provided.

5. In the Discussion, the authors state that “other peptide masses were found to be different from those in previous studies, which can be explained by the different preparation protocols…”. This is of concern if the purpose of the study is to develop a new, reproducible method for serotype detection. Some different preparation protocols should be considered and examined with the current study, to attempt to provide common spectra that could be used by other groups and their own protocols.

6. The authors state in the Discussion that “the obtained PMFs dendrogram in this study (Fig 2) clustered both SS32 and SS34 together but separate from the other S. suis isolates, indicating that these two serotypes possibly possessed a high genetic dissimilarity from the other examined SSs”. Provide a dendrogram based on 16S RNA to show this, and any dissimilarity of other serotypes. Make a comparison between this and the dendrogram for the mass spectrometry data.

7. Within the Discussion, the authors state that “In order to improve the correct serotype classification rate, we suggested that the repeatability test of the serotype classification, including unknown serotyping, should require at least two times more than 8 MS spectra (spots)”. This reviewer recommends that the authors carry this out.

8. The complete list of m/z identified and used should be provided as a data supplement.

Reviewer #2: This manuscript describes a novel matrix-assisted laser desorption/ionization time-of-flight mass spectrometry (MALDI-TOF-MS) analysis of the extracted peptides obtained from Streptococcus suis, and in particularly S. suis serotype (SS2). In this method, cellular proteins (and peptides) of S. suis are extracted and then used for species identification via peptide mass fingerprints (PMFs) formed using MALDI-TOF-MS.

The authors showed that this MALDI-TOF-MS analysis permitted to improve the classification of 32 serotypes of S. suis. Indeed, each individual MALDI-TOF-MS exhibited an individual PMFs patterns, which allowed to differentiate each serotype. This was followed by developing an exclusive peptide mass fingerprint (PMFs) database of S. suis, which was generated from the whole-cell peptides of 32 reference strains of S. suis. This PMFs database was generated by inserting 20 qualified MS spectra from each of the 32 individual reference serotypes into the MALDI Biotyper database system according to Bruker’s recommendation.

It should be noted that although, it was suggested that multiplex (m)PCR analysis could be used to serotype S. suis isolates. However, it was shown that it did not allow the differentiation of SS2 from SS1/2, or SS14 from SS1 due to the high capsular gene cluster similarity. The authors showed that this discrepancy could be resolved by using the PMFs blueprints to discriminate the previously indistinguishable highly pathogenic SS2 and SS14 from SS1/2 and SS1. Indeed, serotyping using MALDI-TOF-MS correctly classified SS2 from SS1/2, or SS14 from SS1serotypes in 68.8% (22/32); while the validity for the clinical human isolates was 62.5% (20/32). The agreement between the MALDI-TOF MS and mPCR serotyping was moderate with a Kappa score of 0.522, considering that mPCR could correctly serotype up to 75%.

In conclusion, the authors have demonstrated that PMFs from the developed MALDI-TOF-MS based method can successfully discriminate the previously indistinguishable highly pathogenic SS2 and SS14 from SS1/2 and SS1, respectively. Moreover, this serotyping method distinguished pathogenic SS6, and so is an alternative approach of choice to rapidly and reliably serotype clinically pathogenic S. suis isolates.

This manuscript is generally well written and the authors have shown that by using MALDI-TOF-MS serotyping using PMFs it was possible to classify the correct serotypes of S. suis isolates.

Nevertheless, few changes are requested to improve the scientific flow of this manuscript.

1. To be compliant with the IUPAC MS nomenclature, please change all your MALDI-TOF MS into MALDI-TOF-MS.

2. Your Figure 2 is illegible, change for a better and crisper one.

3. I am confused and cannot understand the following argument of page 14:

Cohen’s unweighted Kappa statistic was used to elucidate the agreement between the mPCR, as the gold standard method, and this developed MALDI-TOF MS serotyping method, and gave a Kappa score of approximately 0.522. While the true serotype classifications of the mPCR and MALDI-TOF MS, based on the results from Table 4, were approximately 76% (78/103) and 60% (62/103), respectively

Query: Is the serotyping using mPCR (76 %) better than the MALDI-TOF-MS (60%)? Then I suggest that for the validation of your manuscript, you make it clearer. You must stress the benefit of your PMFs method allows better discrimination between the serotypes especially for the previously indistinguishable highly pathogenic SS2 and SS14 from SS1/2 and SS1, respectively

6. PLOS authors have the option to publish the peer review history of their article (what does this mean?). If published, this will include your full peer review and any attached files.

Reviewer #1: No

Reviewer #2: No

---

## [Author Response · Author response to Decision Letter 0]

7 Mar 2021

Response to reviewers of PLOS ONE

We would like to thank the editor and the reviewers for their thoughtful comments. We have carefully addressed all the comments. The corresponding changes and refinements made in the revised paper are summarized in our response below. 

Reviewer#1: In the manuscript titled “Streptococcus suis serotyping by matrix-assisted laser desorption/ionization time-of-flight mass spectrometry”, Chaiden et al. used MALDI-TOF mass spectrometry to serotype Streptococcus suis from reference samples and compare against the use of multiplex-PCR, then validate the mass spectrometry method using clinical isolates. Specific comments are as follows:

1. On line 83, it is a bit misleading to state that “the classification of SSs has never been reported”. In reading through reference #8, it seems the authors of that paper have been able to examine serotypes 2, 7, and 9, as well as what they referred to as “less prevalent” serotypes. Though not a complete listing of serotypes (as it seems the authors assessed at least eight serotypes), the authors of the current manuscript cannot state that any classification/serotype attempts have not been reported previously. This needs to be corrected through the manuscript, where appropriate.

RESPONSE : We thank the reviewer for pointing this out. After considering, we decided to remove the phrase “never been reported” to “Nevertheless, one previous study successfully examined few serotypes of SS using MALDI-TOF-MS still most of the serotypes were not classified especially undistinguishable serotypes by current serotyping methods [8].”

2. Is multiplex-PCR really considered a “gold standard”, when it has a reasonably poor rate of identifying serotypes? Sequencing, and likely of just the 16S RNA, would really provide near 100% serotype identity. (After all, the authors in the current manuscript indicate that the 32 reference strains were determined by sequencing of the 16S rRNA). It is strongly recommended to remove reference to “gold standard” throughout the manuscript regarding multiplex-PCR.

RESPONSE : The reviewer has made some excellent points and we sincerely appreciated these well-thought comments. However, sequencing of the 16S rRNA gene was performed only to confirm the reference strains used in this study. As our objective was to determine the ability of serotyping methods commonly used by the laboratories, thus mPCR detecting cps gene was chosen for agreement testing between two serotyping methods. Nevertheless, we recognized this insightful suggestion from reviewer, so reference to “gold standard” throughout the manuscript regarding mPCR was removed. Moreover, to make it clearer, the mPCR results in Table 2-3 were also removed. 

3. Under “Validation”, provide the source/location of the clinical isolates used (i.e., which hospital or clinic, and what parts of the country the subjects came from).

RESPONSE : We agreed and have updated the source/location of the clinical isolates used in materials and methods section line 101, “local S. suis SS2 (n = 23) and SS14 (n = 9) isolates from humans were used, the strains were originally collected from hospital, distributed in 5 regions (north, northeast, central, east, and south) of Thailand.”

4. Figure 2 is not clear at all at the resolution provided.

RESPONSE : We apologized for our original Figure 2. We have modified the figure and hope that it is now clear.

5. In the Discussion, the authors state that “other peptide masses were found to be different from those in previous studies, which can be explained by the different preparation protocols…”. This is of concern if the purpose of the study is to develop a new, reproducible method for serotype detection. Some different preparation protocols should be considered and examined with the current study, to attempt to provide common spectra that could be used by other groups and their own protocols.

RESPONSE : We agreed with the reviewer that further elaborating on this point would be helpful. However, the peptide extraction method was only modified from Bruker’s recommendation, since it could not provide satisfying signals for serotyping. We also pointed out some different preparation protocols to Paragraph 2 in Discussion “In the present study, the principal difference in the preparation protocol was the modified peptides extraction method, that formic acid was not included, gentle dissolve pellet by vortex and vigorously mix peptides with sinapinic matrix, which likely accounted for some of the dissimilar PMFs. Moreover, dissimilar PMFs could be inherent in the intraspecies discrepancy as well. 

6. The authors state in the Discussion that “the obtained PMFs dendrogram in this study (Fig 2) clustered both SS32 and SS34 together but separate from the other S. suis isolates, indicating that these two serotypes possibly possessed a high genetic dissimilarity from the other examined SSs”. Provide a dendrogram based on 16S RNA to show this, and any dissimilarity of other serotypes. Make a comparison between this and the dendrogram for the mass spectrometry data.

RESPONSE : We agreed and have updated a phylogenetic tree base on 16s rRNA gene as Figure 4. We also made a comparison to Paragraph 3 in Discussion “The phylogenetic tree based on 16s rRNA gene (Fig 4) also supported this finding as SS32 and SS34 were also clustered separately from the main group of S. suis”, to Paragraph 3 in Discussion “However, these divergent characters were not displayed in phylogenetic tree (Fig 4)” and to Paragraph 4 in Discussion “Notably that, these two pairs of serotypes closely clustered together in phylogenetic tree (Fig 4) but separately in peptide dendrogram (Fig 2). This finding could be used to explain why our modified method could differentiate these two pairs of high similarity serotypes”

7. Within the Discussion, the authors state that “In order to improve the correct serotype classification rate, we suggested that the repeatability test of the serotype classification, including unknown serotyping, should require at least two times more than 8 MS spectra (spots)”. This reviewer recommends that the authors carry this out.

RESPONSE : We have fixed the error by changing the word “more than” to “at least” to Paragraph 5 in Discussion.

8. The complete list of m/z identified and used should be provided as a data supplement.

RESPONSE : We agreed and have submitted the complete list of m/z identified as a supplement data.

Reviewer#2: This manuscript describes a novel matrix-assisted laser desorption/ionization time-of-flight mass spectrometry (MALDI-TOF-MS) analysis of the extracted peptides obtained from Streptococcus suis, and in particularly S. suis serotype (SS2). In this method, cellular proteins (and peptides) of S. suis are extracted and then used for species identification via peptide mass fingerprints (PMFs) formed using MALDI-TOF-MS.

The authors showed that this MALDI-TOF-MS analysis permitted to improve the classification of 32 serotypes of S. suis. Indeed, each individual MALDI-TOF-MS exhibited an individual PMFs patterns, which allowed to differentiate each serotype. This was followed by developing an exclusive peptide mass fingerprint (PMFs) database of S. suis, which was generated from the whole-cell peptides of 32 reference strains of S. suis. This PMFs database was generated by inserting 20 qualified MS spectra from each of the 32 individual reference serotypes into the MALDI Biotyper database system according to Bruker’s recommendation. It should be noted that although, it was suggested that multiplex (m)PCR analysis could be used to serotype S. suis isolates. However, it was shown that it did not allow the differentiation of SS2 from SS1/2, or SS14 from SS1 due to the high capsular gene cluster similarity. The authors showed that this discrepancy could be resolved by using the PMFs blueprints to discriminate the previously indistinguishable highly pathogenic SS2 and SS14 from SS1/2 and SS1. Indeed, serotyping using MALDI-TOF-MS correctly classified SS2 from SS1/2, or SS14 from SS1serotypes in 68.8% (22/32); while the validity for the clinical human isolates was 62.5% (20/32). The agreement between the MALDI-TOF MS and mPCR serotyping was moderate with a Kappa score of 0.522, considering that mPCR could correctly serotype up to 75%. In conclusion, the authors have demonstrated that PMFs from the developed MALDI-TOF-MS based method can successfully discriminate the previously indistinguishable highly pathogenic SS2 and SS14 from SS1/2 and SS1, respectively. Moreover, this serotyping method distinguished pathogenic SS6, and so is an alternative approach of choice to rapidly and reliably serotype clinically pathogenic S. suis isolates. This manuscript is generally well written and the authors have shown that by using MALDI-TOF-MS serotyping using PMFs it was possible to classify the correct serotypes of S. suis isolates. Nevertheless, few changes are requested to improve the scientific flow of this manuscript.

1. To be compliant with the IUPAC MS nomenclature, please change all your MALDI-TOF MS into MALDI-TOF-MS.

RESPONSE : We have fixed the error by changing the words from MALDI-TOF MS into MALDI-TOF-MS

2. Your Figure 2 is illegible, change for a better and crisper one.

RESPONSE : We apologize for our original Fig 2. We have modified the figure and hope that it is now clear.

3. I am confused and cannot understand the following argument of page 14:

Cohen’s unweighted Kappa statistic was used to elucidate the agreement between the mPCR, as the gold standard method, and this developed MALDI-TOF MS serotyping method, and gave a Kappa score of approximately 0.522. While the true serotype classifications of the mPCR and MALDI-TOF MS, based on the results from Table 4, were approximately 76% (78/103) and 60% (62/103), respectively

Query: Is the serotyping using mPCR (76 %) better than the MALDI-TOF-MS (60%)? Then I suggest that for the validation of your manuscript, you make it clearer. You must stress the benefit of your PMFs method allows better discrimination between the serotypes especially for the previously indistinguishable highly pathogenic SS2 and SS14 from SS1/2 and SS1, respectively

RESPONSE : We recognized this insightful suggestion from reviewer, thus to Paragraph 8 in Discussion these following sentences were added “this MALDI-TOF-MS serotyping method allows better discrimination between the serotypes especially for the previously indistinguishable highly pathogenic SS2 and SS14 from SS1/2 and SS1, respectively.” To make it clear, we also decided to remove the word “gold standard” and mPCR results in Table 2. Moreover, we also add 95% CI value of both methods in line 258 “While the true serotype classifications of the MALDI-TOF-MS and mPCR were not statistically significant (p > 0.05) approximately 60% (95% CI: 51%-70%) and 76% (95% CI: 67%-84%), respectively.” We also discussed more in line 364 “Nevertheless, 95% CI of both methods indicated that the true serotype classification rate of both methods was not statistically different”.

---

## [Decision Letter · Decision Letter 1]

23 Mar 2021

Streptococcus suis serotyp ing by matrix-assisted laser desorption/ionization time-of-flight mass spectrometry

PONE-D-21-02524R1

Dear Dr. Nuanualsuwan,

We’re pleased to inform you that your manuscript has been judged scientifically suitable for publication and will be formally accepted for publication once it meets all outstanding technical requirements.

Kind regards,

Joseph Banoub, Ph,D., D. Sc.

Academic Editor

PLOS ONE

Additional Editor Comments (optional):

The authors have answered all the demanded queries of both referees.

This manuscript is now acceptable.

Reviewers' comments:

Reviewer's Responses to Questions

**Comments to the Author**

1. If the authors have adequately addressed your comments raised in a previous round of review and you feel that this manuscript is now acceptable for publication, you may indicate that here to bypass the “Comments to the Author” section, enter your conflict of interest statement in the “Confidential to Editor” section, and submit your "Accept" recommendation.

Reviewer #2: All comments have been addressed

2. Is the manuscript technically sound, and do the data support the conclusions?

Reviewer #2: Yes

3. Has the statistical analysis been performed appropriately and rigorously? 

Reviewer #2: Yes

4. Have the authors made all data underlying the findings in their manuscript fully available?

Reviewer #2: Yes

5. Is the manuscript presented in an intelligible fashion and written in standard English?

Reviewer #2: Yes

6. Review Comments to the Author

Reviewer #2: This is my second review . I noticed that answers from authors to my previous comments bring the required elemnts. I have not more comments.

7. PLOS authors have the option to publish the peer review history of their article (what does this mean?). If published, this will include your full peer review and any attached files.

Reviewer #2: No

---

## [Editor Report · Acceptance letter]

23 Apr 2021

PONE-D-21-02524R1 

*Streptococcus suis* serotyping by matrix-assisted laser desorption/ionization time-of-flight mass spectrometry 

Dear Dr. Nuanualsuwan:

I'm pleased to inform you that your manuscript has been deemed suitable for publication in PLOS ONE. Congratulations! Your manuscript is now with our production department. 

Kind regards, 

on behalf of

Dr. Joseph Banoub 

Academic Editor

PLOS ONE